# The Use of Solid Sodium Silicate as Activator for an Amorphous Wollastonitic Hydraulic Binder

**DOI:** 10.3390/ma17030626

**Published:** 2024-01-27

**Authors:** Mónica Antunes, Rodrigo Lino Santos, João Pereira, Ricardo Bayão Horta, Rogério Colaço

**Affiliations:** 1Intituto Superior Técnico, University of Lisbon, Av. Rovisco Pais, 1049-001 Lisboa, Portugal; monica.h.antunes@tecnico.ulisboa.pt (M.A.);; 2IDMEC—Instituto de Engenharia Mecânica, University of Lisbon, 1649-004 Lisboa, Portugal; 3CIMPOR—Cimentos de Portugal, SGPSS.A., Praceta Teófilo Araújo Rato, 2600-540 Alhandra, Portugal; rlsantos@cimpor.com (R.L.S.);

**Keywords:** cement, alternative binder, alkaline activator, sodium silicate synthesis

## Abstract

To ensure the acceptable mechanical strength of amorphous wollastonitic hydraulic binders (AWHs), activation with a sodium silicate solution is necessary. However, the use of this type of activator increases the final cost and the complexity of the product’s overall use. In this work, we focus on enhancing the manufacturing of the alkaline activator by producing three Na_2_SiO_3_ powders using cost-effective raw materials. The procedure consisted of heating a mixture of NaOH pebbles with either sand, glass, or diatomite to a temperature of 330 °C for 2 h. After synthesis, the powders were characterized by Fourier Transform Infrared Spectroscopy (FTIR) and X-ray Diffraction (XRD) techniques. Finally, mortars made with AWHs were activated using the synthesized powders that were added either as a solid or dissolved in an aqueous solution. The compressive strength results in these mortars show that the lab-made activators are competitive with the traditional sodium silicate activators. Furthermore, the synthetized activators can be added in either solid form or pre-dissolved in a solution. This innovative approach represents a more economical, sustainable and easy-to-use approach to enhancing the competitiveness of AWHs.

## 1. Introduction

The carbon dioxide emitted in the production of Ordinary Portland Cement (OPC) is responsible for a significant part of global CO_2_ emissions [1]. An alternative that has been studied since the 1960s is the substitution of OPC for Alkaline Activated Cements (AACs) [2,3], which can reduce CO_2_ emissions by up to 55–75% [2,3,4,5]. AACs are formed in an alkaline environment that promotes the dissolution of the aluminosilicate precursors, which can originate from either natural or industrial waste materials [5,6]. Glukhovsky et al. [7] classified the alkaline activators into the following six categories, in which M represents an alkali group:Alkalis, MOH;Weak acid salts, M_2_CO_3_, M_2_SO_3_, M_3_PO_4_, and MF;Silicates, M_2_O.*n*SiO_3_;Aluminates, M_2_O.Al_2_O_3_;Aluminosilicates, M_2_O.Al_2_O_3_. (2–6) SiO_2_;Strong acid salts, M_2_SO_4_.

The activation process involves breaking the covalent bonds Si–O–Si and Al–O–Si due to the alkaline pH of the solution, leading to the formation of a colloidal phase, followed by coagulation, which eventually results in a condensed structure [2]. Research on these binders has shown not only good mechanical and durability properties, but also low heat of hydration, high impermeability, and resistance to extreme temperatures, sulfates, sea spray and acid attacks [8]. These promising features make them attractive for various applications.

Previous studies showed that sodium silicate (Na_2_SiO_3_) solutions are efficient activators for amorphous wollastonitic hydraulic binders (AWHs) [9]. However, the use of Na_2_SiO_3_ can be challenging due to its corrosiveness [10] and significant environmental impact caused by its high embodied energy [11,12]. Hence, the production of a green, low-cost, and easy-to-use activator is necessary. One promising approach involves using amorphous soda-lime glass as a source of silica.

Several studies have focused on the synthesis of this activator [13,14] using different silica-rich sources as raw materials. These sources include low-value, silicate-rich wastes [15,16,17,18], and bio-derived materials [19,20]. There are two main routes of synthesis that can be applied [21], a thermochemical method and a hydrothermal method. The thermochemical method consists of mixing the silica-rich powder with a sodium hydroxide powder using a temperature of 450–600 °C [14,19,21], and the hydrothermal method consists of dissolving the silica-rich materials in an alkaline solution at a temperature ≤ 100 °C [15,16,17,18]. Therefore, products produced using the hydrothermal method are liquid and require less energy for production [21]. In both cases, the concentration of the alkaline material [21], the particle size distribution, and the amorphous content of the silica source influence the reaction process [14], since smaller particles have higher SiO_2_ extraction capabilities and amorphous phases display higher solubility [14]. Finally, to analyze the formation of condensed and uncondensed silica [18] and to confirm the characteristic bands associated with the sodium silicate spectra [17], the final product can be characterized by FTIR.

The different routes and raw materials used in the production of the activator may affect its application with the binder, since these factors influence the rate of silicate release in the system [14], and consequently the hydration kinetics, compositions, crystalline phases, and mechanical strength of the AAC [14].

Sasui et al. [11] synthesized an alkaline activator by dissolving waste glass powder in a 4 M NaOH solution and successfully enhanced the properties of class-C fly ash and ground granulated blast furnace slag. Moreover, they also reported that by increasing the amount of dissolved waste glass powder in the alkaline solution, a higher silicon concentration and increased reactivity were observed.

According to Toniolo et al. [22], increasing the amount of waste glass and the molarity of the NaOH solution leads to the formation of zeolite crystalline phases, which in turn enhances the mechanical strength of the final product. However, when using a solution with high NaOH content, it is necessary to balance between the reactivity and structural integrity of the hydration product, since, even though high values (e.g., 10 M) accelerate the dissolution rate of the aluminosilicate precursors, they also promote the disintegration of the network structure after 28 days of curing.

Another innovative approach was proposed by Vinai et al. [23]. These authors produced a sodium silicate powder by mixing glass powder, sodium hydroxide powder and water. This mixture was heated at temperatures ranging from 150 to 330 °C for 2–4 h. The results show that the binder hydrated with the lab-made activator obtained similar compressive strength results to those binders activated with commercial solutions. This represents a significant cost reduction in activator production, depending essentially on the cost of the NaOH, which accounts for 95% of the activator cost [23].

In this study, we focused on the synthesis of a Na_2_SiO_3_ powder using various raw materials. To this end, the method proposed by Vinai et al. [23] was implemented, while also exploring alternative raw material sources such as sand and diatomite as replacements for glass powder. The synthesized powders were analyzed by Fourier Transform Infrared Spectroscopy (FTIR) and X-ray Diffraction (XRD). The reactivity of the powders was assessed by isothermal calorimetry in activated pastes made with the AWH binders. The synthesized powders were introduced either as a solid or in a solution. Finally, the performance of the activators, both as a solid and in solution, was tested in AWH-based mortars by measuring their compressive strength after 2, 7, and 28 days. This work aims to present a cost-effective and environmentally friendly alternative for the production and application of Na_2_SiO_3_ alkaline activators, and to enable the activation of the AWH binder using a more affordable, environmentally sustainable, and user-friendly approach.

## 2. Materials and Methods

### 2.1. Production of the Binder

The AWH binder was made using common cement clinker raw materials, with adjustments in the overall chemical composition to obtain a total C/S molar ratio of 1.05. The raw materials were ground, mixed, and compressed into a disc, which was then broken into four pieces and introduced into a silicon carbide crucible. The filled crucible was first heated in an electric furnace to 900 °C, using a rate of 25 °C/m, and in order to allow chamber to saturate with CO_2_, this temperature was maintained for 1 h. Then, the temperature was increased to 1550 °C and kept for 1 h to ensure the complete melting of the mix and its chemical homogenization. Finally, the molten material was quenched into a water container. The final product was characterized by XRD analysis, and it displayed an amorphous content of 91.8% and 8% of pseudowollastonite. In Table 1, the chemical composition of the produced clinker is displayed. The AWH clinker was then ground with a ring mill for 3 min, obtaining a final Blaine fineness around 4580 cm^2^/g.

### 2.2. Production of the Alkaline Activator

Three activators were produced using either glass, sand or diatomite as a source of silica.

Recycled glass cullet was crushed in a jaw crusher, followed by 3 min of milling in a ring mill. Regarding the sand, AFNOR sand was milled for 3 min in a ring mill, while diatomaceous earth from Rio Maior, Portugal was first crushed in a jaw crusher, followed by 3 min of milling in a ring mill.

To all silica-rich raw materials, commercial-grade NaOH pebbles (98.2% Prolabo, Matsonford, PA, USA) and 10 mL deionized water were added to obtain a final product with a Si:Na ratio of 1 and to facilitate the mixture of the raw materials. The mixture was then stirred with a laboratory rod until a homogeneous paste was obtained. To prevent overflowing, the samples were introduced into the furnace at room temperature and then heated to 330 °C and left for 2 h. After removing from the furnace, the material was allowed to cool to room temperature. Before being used with the binder, the solid activators produced were manually milled.

The chemical composition of the raw materials used in the production of each activator is shown in Table 2. Figure 1 depicts each of the mixtures before and after the heating process.

### 2.3. Analysis of the Activators: Fourier Transform Infrared Spectroscopy (FTIR)

The produced activators were analyzed using FTIR-ATR and X-ray diffraction tests and compared to a standard anhydrous sodium silicate (EastChem, Qingdao, China). The FTIR tests were performed on a Bruker 400 MHz, model ALPHA, operating with a Platinum ATR module, with 4 cm^−1^ resolution over 24 scans.

### 2.4. Analysis of the Activators: X-ray Diffraction (XRD) Analysis

The XRD analysis was performed using an X’Pert Pro (PANalytical, Overijssel Netherlands) diffractometer with monochromatic CuKα1 radiation (λ = 1.54059 Å) and working in reflection geometry (θ/2θ). The optics configuration was a fixed divergence slit (1/2°), a fixed incident anti-scatter slit (1°), a fixed diffracted anti-scatter slit (1/2°), and X’Celerator RTMS (Realtime Multiple Strip) detectors, working in scanning mode with the maximum active length. Data for each sample were collected from 5° to 70° (2θ). The samples were rotated during data collection at 16 rpm to enhance particle statistics. The X-ray tube worked at 45 kV and 40 mA.

### 2.5. Production of the Activating Solution

For this study, we prepared three types of solutions, with a SiO_2_/Na_2_O modulus (MS) of 1 and a Na_2_O molarity of 3.516 M. The first solution was a standard sodium silicate solution, obtained by mixing Na_2_SiO_3_ (Na_2_O: 7.5–8.5%, SiO_2_: 25.5–28.5%, Chem Lab, Zedelgem, Belgium) with NaOH (98.2%, Prolabo, Matsonford, PA, USA). The second solution was made using anhydrous sodium silicate (EastChem, Qingdao, China) combined with solid NaOH (98.2%, Prolabo, Matsonford, PA, USA); this mixture was then dissolved in deionized water. Lastly, the third type of solution was made by dissolving the lab-made activator powder into deionized water.

### 2.6. Production of Pastes

Pastes were prepared with a water/solid ratio of 0.25. The lab-made activators were introduced in two ways: either as a solid directly mixed with the anhydrous binder or pre-dissolved into a solution. Then, to the aqueous solution the solids were introduced and mixed for 90 s using a bench-top mixture. The resulting pastes were molded into inox steel molds with 20 × 20 × 40 mm^3^ dimensions and cured at 20 °C under relative humidity (HR) conditions above HR95% during the entire experiment. The samples were demolded after 2 days and left to cure under the same conditions. Finally, one compressive strength test per age was performed at 2, 7, and 28 days of hydration in an Ibertest Autotest 400/10 instrument using a constant force rate of 2.4 kN/s during the test.

### 2.7. Isothermal Calorimetry

For the isothermal calorimetry tests, pastes made with the AWH binder, and with a water/solid ratio of 0.25, were introduced into vials via ex situ in an A TAM Air instrument (Waters Sverige AB, Sollentuna, Sweden). The tests were run at a constant temperature of 20 °C. 

### 2.8. Production of Mortars

Mortars were prepared by mixing the binder with the solution using a water/solid ratio of 0.365, and by adding 1350 g of AFNOR sand. The lab-made activators were introduced in two ways: either as a solid, directly mixed with the anhydrous binder, or pre-dissolved into the solution. The mixing was performed in a Tonimix mixer from ToniTechnik for 120 s; the resulting mortar was introduced in 40 × 40 × 160 mm^3^ inox steel molds and left to cure at 20 °C under HR conditions above HR95% during the entire experiment. The samples were demolded after 1 day and left to cure under the same conditions, and one replica per age was teste. Finally, one replica per age was tested, which included one flexural test and two compressive strength tests after 2, 7, and 28 days of hydration in an Ibertest Autotest 400/10 instrument. For comparison purposes, a mortar made with a standard sodium silicate solution was also prepared.

### 2.9. Thermogravimetric Analysis (TGA)

The material obtained from the compressive strength test was manually ground and dried at 105 °C to eliminate any residual evaporable water. Then, to calculate the amount of bound water (BW) in the hydrated phases, thermogravimetry analysis was performed by calculating the weight loss of the pastes and mortar over time at specific temperature intervals. The analysis was made in an ELTRA multichannel TGA device, with constant heating rates at fixed temperatures of 105, 250, 500, and 950 °C until a stable mass was reached. The heating rates varied: 4 °C/min from room temperature to 105 °C, 10 °C/min from 105 °C to 250 °C, and 15 °C/min from 250 °C to 500 °C and between 500 °C and 950 °C. Finally, the amount of BW was calculated according to the mass loss between 105 and 500 °C.

## 3. Results

### 3.1. Characterization of the Activators by FTIR

The FTIR results of the lab-made activators are compared with a standard solid sodium silicate (EastChem), as shown in Figure 2.

The principal bands of the FTIR spectra are identified as follows:Broad band between 2500 and 3800 cm^−1^ associated with O–H stretching vibration [24,25];CO_3_^2−^ characteristic bands at 1450 cm^−1^ attributed to asymmetric stretching mode [25];Bands 1050–1100 cm^−1^ characteristic of stretching Si–O–Si bond [20,24,26];Band at 973 cm^−1^ attributed to the Si–O symmetric stretching [25];Band associated with Si–O–Si asymmetric stretching vibration, specifically Q^0^ and Q^1^ at ∼850, ∼900 cm^−1^ [14,27];CO_3_^2−^ characteristic bands at 712 cm^−1^ caused by the in-plane bending modes [28,29,30]. The band at 680 cm^−1^ can be attributed to the bending motion of oxygen bonds [27];Al–OH bending vibrations at ~590–570 cm^−1^ [31] and bands associated to Si–O–Al–O bonds at 459–572 cm^−1^ [16].

The broad band observed between 2500 and 3800 cm^−1^ (A) in the glass and diatomite-based activators suggests either water adsorption [24,25] or a more complex structure, since an increase in these bands is also linked to an increase in OH^−^ groups at the surface of polymerized silica [24]. The results also reveal the presence of CO_3_^2−^ bands in the glass and diatomite-based samples at 1450 and 712 cm^−1^ [25].

The strip highlighted as (D) reveals that all spectra exhibit a band at 950 cm^−1^, indicating the presence of the Na–O–Si bond [20], hence suggesting an effective reaction [20]. In the diatomite-based sample, the band is broader, possibly due to the higher amount of Si–O–T (T = Si, Al) structures (946 cm^−1^) [32].

The doublet at ~890 indicates the presence of Q^0^ structures at 850 cm^−1^ and Q^1^ structures at 900 cm^−1^ [27]. The bands at 875 cm^−1^ and 712 cm^−1^ can be attributed to the in-plane bending mode of CO_3_^2−^ [28,29,30].

All samples exhibit characteristic bands of Al–O bonds at 580 cm^−1^ [31]; however, the diatomite-based sample has a broader band compared to the other spectra, displaying a band at 572 cm^−1^, which can be attributed to Si–O–Al–O bonds [16]. This may be a result of a higher Al content on the raw material of the activator.

### 3.2. Characterization of the Activators by XRD

In Figure 3, the XRDs of the lab-made activators are compared with those of the commercially available EastChem sodium silicate powder. All samples displayed a crystalline structure, and the following nomenclature was given to the identified peaks: 1—Na_6_Si_2_O_7_ [23], 2—Na_2_SiO_3_ [20], 3—CaAl_2_O_4_ [23], 4—CaAl_2_O_4_ [23] and 5—Na_2_CO_3_ [33]. The samples produced with sand and diatomite presented a peak at 26.6° (2θ) (Q), which is indicative of the presence of unreacted silica [16]. Both activators were prepared using silica-rich raw materials with a higher crystalline content than glass, which is predominantly amorphous. This difference may have affected the SiO_2_ solubility and, consequently, the reaction process.

### 3.3. Isothermal Calorimetry Results

The calorimetry results for activated pastes prepared with different activator types, introduced as liquid or solid, are shown in Figure 4, which displays the normalized heat flow and cumulative heat for each of the tested pastes.

The experimental heat evolution curves can be divided into three main stages—a first stage, where a slow kinetic reaction can be observed, followed by an acceleration period, characterized by a high release of heat rate, and finally a deceleration period, where a decrease in the rate of the reaction is observed. Traditionally, in OPC, these stages are related to the dissolution of ionic species, the precipitation of CSH, and the densification of the matrix [9,34,35].

With the exception of the glass-based activator, which already contained absorbed water (see FTIR results), all samples displayed two exothermal peaks. This corresponds to different phenomena, such as the water adsorption by the precursor [36] and the formation of different hydration products beside CSH, such as CASH gel [36], since the diatomite-based activator has a higher Al content. Amongst the lab-made activators, the sand-based samples exhibited the highest cumulative heat released after 2.5 days of hydration, suggesting a more extensive reaction. On the other hand, the diatomite-based sample displayed the lowest cumulative heat released, indicating a lower formation of hydration products. This may be attributed to the higher complexity of the raw materials and a higher level of impurities in the activator.

### 3.4. Compressive Strength Results

#### 3.4.1. Pastes

After 2, 7, and 28 days of hydration, compressive strength tests were performed on the pastes. The results are displayed in Figure 5. At early ages, 2 and 7 days, the compressive strength was similar in all tested samples, ranging from 15 to 23 MPa and 57 to 73 MPa, respectively. Notably, the lowest compressive strength observed at 2 days corresponds to the diatomite-based activated sample, which also registered the lowest cumulative heat, as determined by isothermal calorimetry, suggesting the formation of a lower amount of hydration products at this age.

However, a significant discrepancy is observed in the 28-day results, particularly for the pastes prepared with a solution of sand-based activator and the solid form of diatomite-based activator. In these samples, the compressive strength was 67 and 70 MPa, respectively, which contrasts with the values obtained for the other systems prepared, which achieved compressive strength values of 96–97 MPa in the case of solid-form activators, and over 100 MPa when considering the solution-form activators.

#### 3.4.2. Mortars

After 2, 7, and 28 days of hydration, compressive strength tests were performed on the mortars, and the results are displayed in Figure 6. After 7 and 28 days of curing, all tested samples displayed similar compressive strengths. However, similarly to the paste results, the diatomite-based sample exhibited the lowest value after 2 days of hydration.

### 3.5. TGA

Table 3 displays the experimental TGA data obtained for pastes and mortars tested after 2, 7 and 28 days of curing. It includes the respective loss on ignition (LOI) during the 105–250 °C, 250–500 °C, and 500–950 °C steps. The step from room temperature to 105 °C is not included, as it only relates to the humidity in the sample. To obtain the amount of BW in the sample, only the results at 105–250 °C and 250–500 °C have been considered.

Since the strength development of the samples is directly related to the hydration process, it is possible to relate the compressive strength with the chemical BW. Furthermore, assuming the model of Richardson and Qomi [37,38], a relation between the bound water and the amount of CSH can be established, according to Equation (1).
(1)$$\frac{H}{S}=\frac{19}{17}\times \frac{C}{S}-\frac{7}{17}$$
where the H/S ratio is considered for a tobermorite-like CSH gel structure and a C/S ratio of 1.1 is assumed, resulting in a C_x_SH_y_ approximated structure with x = 1.1 and y = 0.82 [9,39]. Therefore, by considering the chemical composition of the hydrated product and the BW content of each sample, determined by the weight loss between 250 and 500 ºC in the TGA results, the wt. % CSH can be determined.

Using these data, it is possible to plot the weight percentage of CSH against the compressive strength of every sample, as shown in Figure 7 for pastes (blue line) and mortars (orange line); the obtained correlations are displayed in Equations (2) and (3).
(2)$$\mathrm{Compressive}\;\mathrm{strength}\;\mathrm{mortars}\left(MPa\right)=6.7\left(\%CSH\right)-32.1\;R^{2}=0.8$$
(3)$$\mathrm{Compressive}\;\mathrm{strength}\;\mathrm{paste}\left(MPa\right)=4.3\left(\%CSH\right)-72.2\;R^{2}=0.8$$

## 4. Discussion

In this work, different sodium silicate powders were synthesized by mixing NaOH with different silica sources as raw materials. To assess the degree of reaction and understand the composition and structural characteristics of the resulting powders, FTIR and XRD analyses were performed. The XRD analysis showed that the powders have different crystalline phases; the identified peaks can be attributed to sodium silicate with different SiO_2_:Na_2_O ratios, with Na_6_Si_2_O_7_ and Na_2_SiO_3_, calcium aluminum oxide CaAl_2_O_4,_ and sodium carbonate Na_2_CO_3_. These results indicate a successful synthesis of the material; nevertheless, the diatomite- and sand-based activators exhibited a peak at 26.6° (2θ), indicating the presence of unreacted silica in the final product. Furthermore, when analyzing the FTIR spectra on both the glass- and diatomite-based activator, evidence of -OH bonds was found. In the glass-based activator, this band is broader, stretching between 3700 and 3100 cm^−1^, indicative of the hydration of the powder. In the diatomite-based powder, this band is narrower and centers around 3400 cm^−1^, which is characteristic of the OH in NaOH [40], revealing a residual presence of the raw materials and an incomplete reaction.

The lab-made activators were then used to activate the AWH binder. Both were added in solid form and in aqueous solution form. Their reactivity was followed by calorimetric analysis, and the results show that the state at which the activator was added did not significantly influence the hydration reaction. After 2.5 days, all tests indicated a comparable cumulative heat, ranging from 47 to 61 J/g. Nevertheless, the sand- and the glass-based activators presented a slightly higher cumulative heat when added as a solid, while the remaining activators showed better results when introduced as a solution.

The influence of different forms of sodium silicate introductions in the compressive strength is an area of ongoing investigation [36,41]. In this study, the paste’s compressive strength results show that the main difference was observed at 28 days in the pastes prepared with a solution of diatomite-based activator and the solid form of the sand-based activator.

Nevertheless, when observing the amount of CSH formed in these two pastes (blue crosses in Figure 7), the results are similar to those of the other tested samples. Therefore, this underperformance might be related to some defects in the test specimens, since, in both cases, their respective activating conditions (solid or aqueous solution) present compressive strength values within the expected range, and comparable with commercially available activators.

Similarly, Gong et al. [36] also reported that when solid sodium silicates are introduced, mechanical properties comparable to the standard method of activation are obtained. These results also allow one to discard any influence of the activator form from the results obtained in this study, suggesting that both forms can be used without compromising mechanical performance.

Considering all the results from the mortar tests, mean compressive strength values and their respective standard deviations were calculated for each age, and the results are displayed in Table 4. The higher discrepancy at early ages may be linked to variations in the silica release rates of the activators and the differences observed in the kinetic rate in the calorimetric results. Nevertheless, the standard deviations obtained at 7 and 28 days are within the experimental error [42], indicating that at later, ages all samples exhibited similar behaviors.

In conclusion, when tested on mortars, the compressive strength results show that all the lab-made activators were competitive with traditional sodium silicate solution, regardless of the state in which the activator was incorporated. These results also point to the possible defects of paste test specimens that underperformed at 28 days.

Finally, a strong correlation between the amount of CSH formed and the compressive strength was established for both mortars and pastes (Figure 7), strongly suggesting that the mechanical properties of the sample depend on the formation of this hydration product. These results correlate well with those from a previous work [43], wherein mortars were prepared with a similar hydraulic binder under alkaline activation conditions, showing that similar CSH gels were produced in both studies.

Furthermore, by analyzing the trend lines obtained in Figure 7, it is possible to identify the minimum amount of CSH required for the sample to develop compressive strength: ~17 wt. % for pastes and 5 wt. % for mortars. Nevertheless, when compared to previous results for OPC pastes [9,43], we observed that the minimum wt. % CSH required for this material was ~35 wt. %. This low limit for the minimum amount of CSH in the AWH sample may be related to the presence of hydration products with a lower C/S ratio. Previous studies reported that the hydration products of this binder are characterized for having a C/S ratio of 1.1 and have the particularity of producing tobermorite 9 Å [44,45]. This characteristic allows the formation of a CSH structure with improved mechanical properties, specifically in terms of hardness and stiffness [38,44].

## 5. Conclusions

In this work, different lab-made Na_2_SiO_3_ powders were synthesized by using low-cost raw materials, such as diatomite, sand and glass waste. FTIR and XRD analysis confirmed the successful synthesis of these materials. Each synthesized powder was used to activate pastes, and the calorimetric analysis revealed comparable reactivity among the activators, whether added as solids or pre-dissolved into solutions. Furthermore, the compressive strength tests of pastes and mortars demonstrated that the lab-made activators are competitive with traditional sodium silicate solutions, regardless of the form in which they are added. Finally, the CSH gels formed from the activation of the AWH binder with the lab-made activators presented similarities with previous studies on the activation of AWH with commercial activators, where a strong correlation between the amount of CSH formed and the compressive strength for both mortars and pastes was observed. Furthermore, the minimum amount of CSH required for the AWH to develop compressive strength was determined to be 17 wt. % for pastes and 5 wt. % for mortars. This result is lower than the amount of OPC required, which may be due to the presence of hydration products with a lower C/S ratio in the AWH binder. In conclusion, the synthesis of the activator presented in this paper may represent a cheaper, more environmentally sustainable and easy-to-use alternative for the production and application of alkaline activators.

## Figures and Tables

**Figure 1 materials-17-00626-f001:**
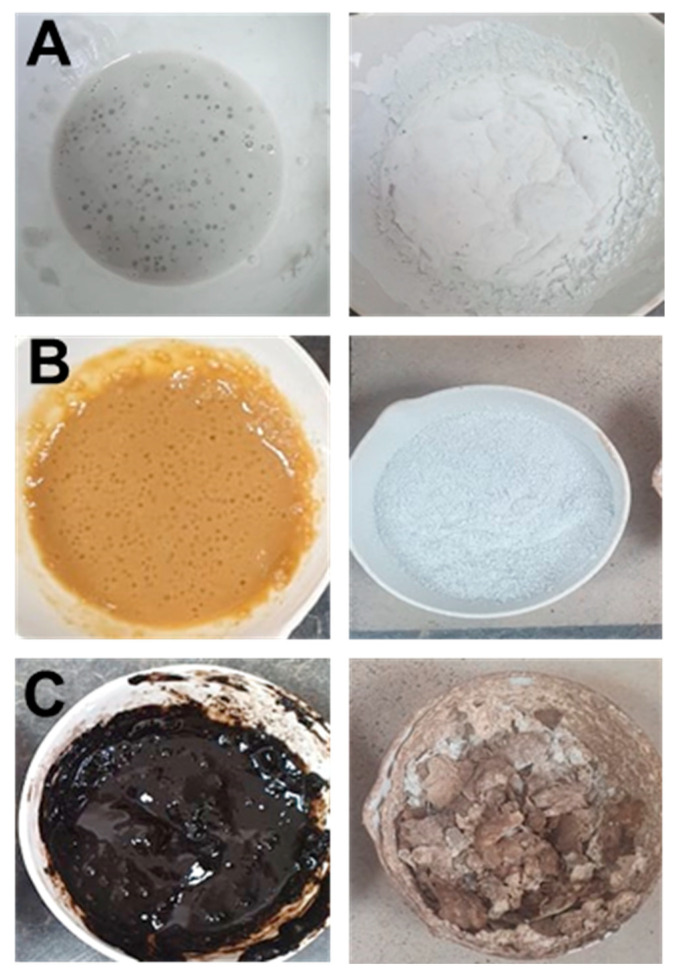
Lab-made activator before and after the heating process. (**A**) Glass based activator, (**B**) sand-based activation, (**C**) diatomite-based activator.

**Figure 2 materials-17-00626-f002:**
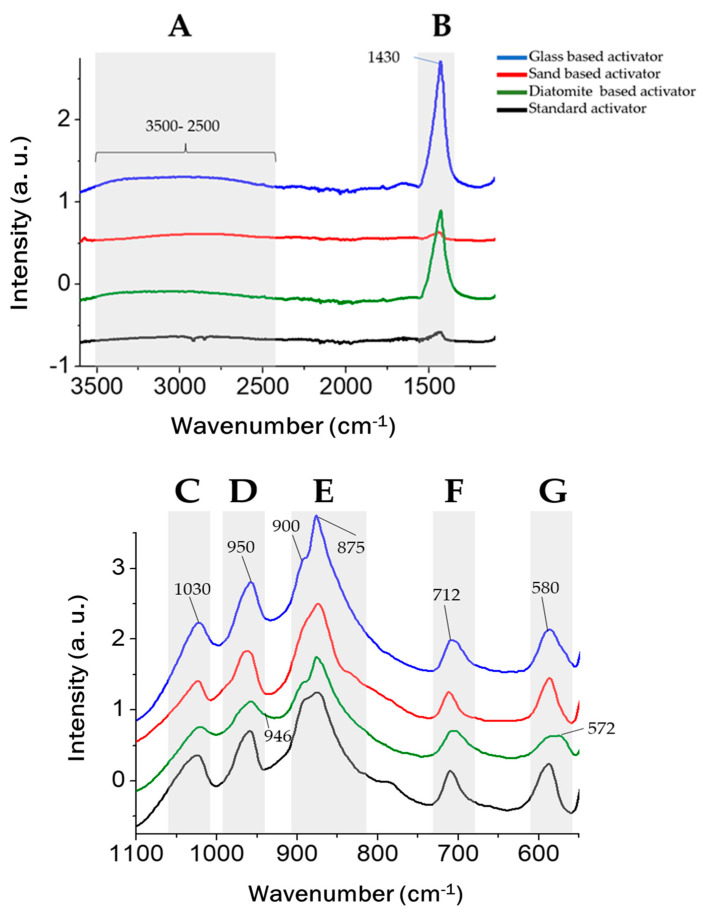
Comparison of the FTIR results of the synthesized activators with a standard, commercially available EastChem powder.

**Figure 3 materials-17-00626-f003:**
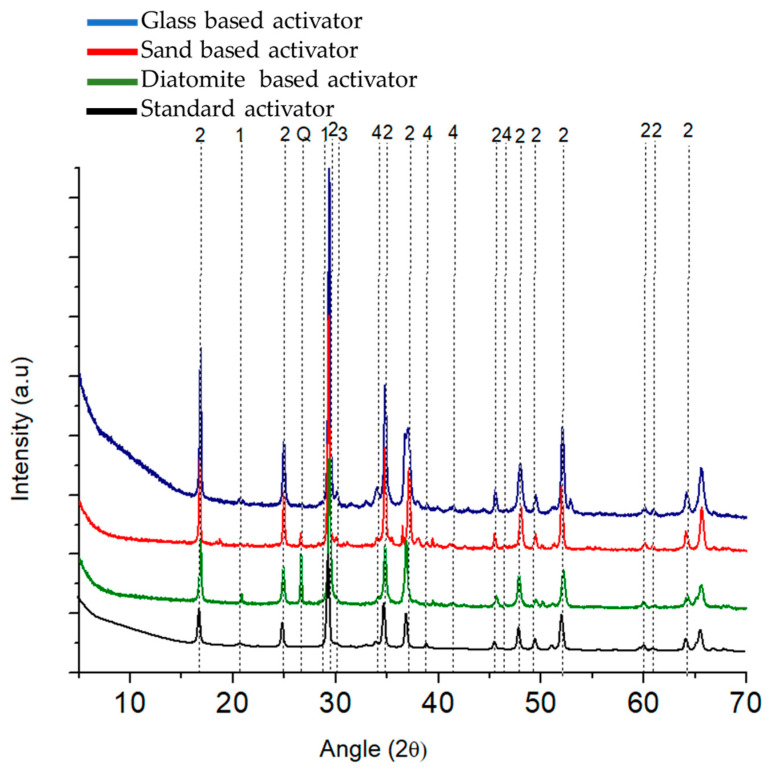
Comparison of the XRD results of the synthesized activators with a powder bought from EastChem. 1—Na_6_Si_2_O_7_, 2—Na_2_SiO_3_, 3—CaAl_2_O_4,_ and 4—Na_2_CO_3_. Q—unreacted silica.

**Figure 4 materials-17-00626-f004:**
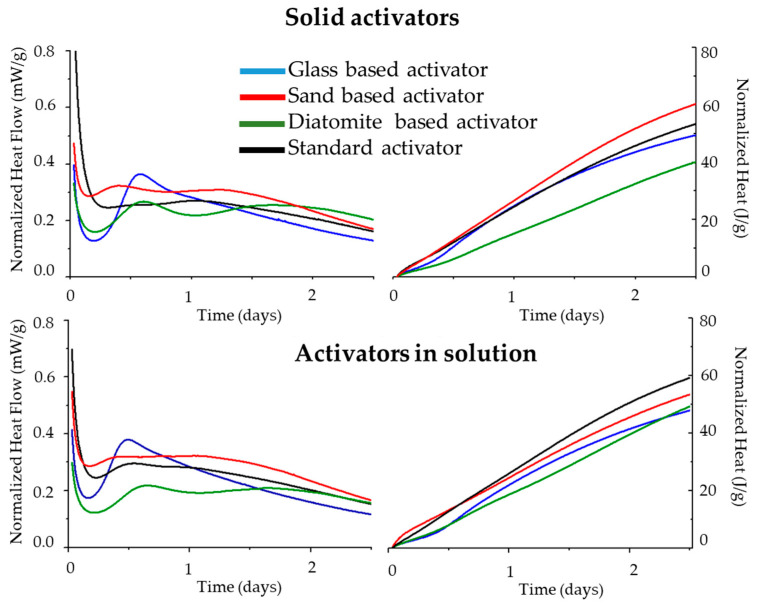
Isothermal calorimetry results of the AWH pastes hydrated with sodium silicate activators produced with different raw materials. Activators were added either as a solid (**top row**) or as a solution (**down row**).

**Figure 5 materials-17-00626-f005:**
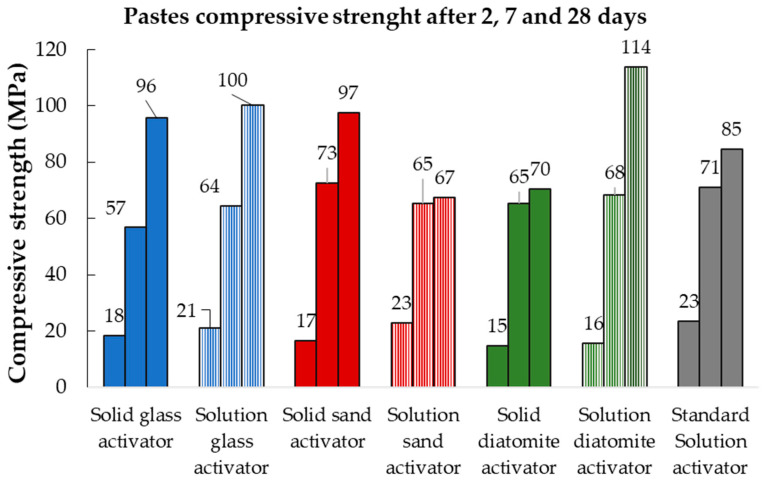
Paste compressive strength results, activated either with the lab-made activators or with the commercial solution.

**Figure 6 materials-17-00626-f006:**
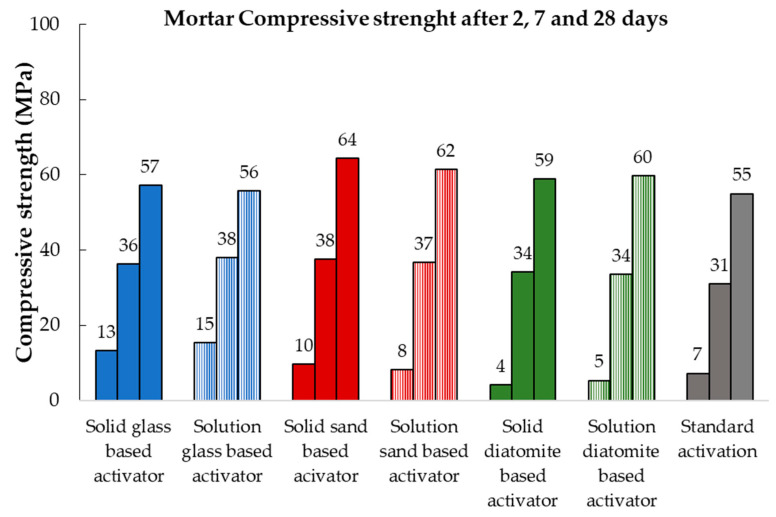
Comparison of the compressive strength results of the mortar activated using the standard alkaline solution (grey bars) with those of the mortars activated using the lab-made activators.

**Figure 7 materials-17-00626-f007:**
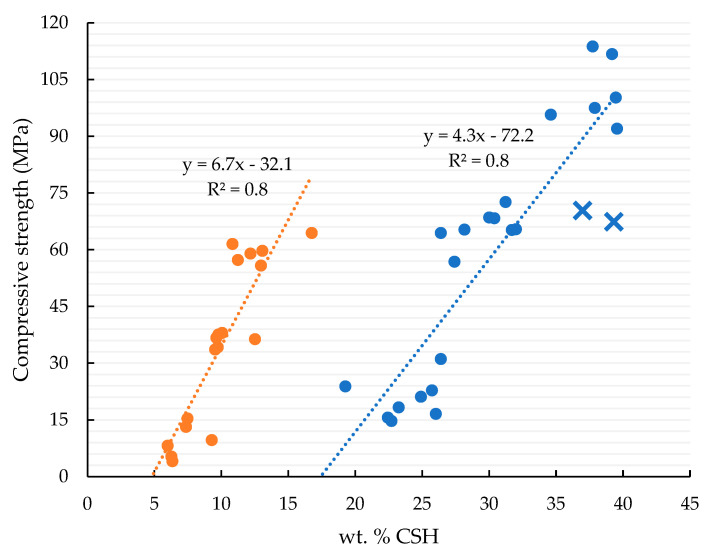
Correlation between the wt. % CSH formed and the compressive strength, for both pastes (blue line) and mortars (orange line). The blue crosses represent the results for the 28-day hydrated pastes that underperformed in the compressive strength tests, the solid diatomite-based and solution sand-based activated pastes.

**Table 1 materials-17-00626-t001:** Chemical composition of the AWH clinker produced, wt. %.

wt. %	SiO_2_	Al_2_O_3_	Fe_2_O_3_	CaO	MgO	SO_3_	K_2_O	Na_2_O	Others
AWH	48.73	2.05	0.71	47.42	0.32	0.04	0.33	0.12	0.05

**Table 2 materials-17-00626-t002:** Composition of the raw material used in the production of the activator, wt. %.

wt. %	SiO_2_	Al_2_O_3_	Fe_2_O_3_	CaO	MgO	SO_3_	K_2_O	Na_2_O	Others
Glass	72.39	1.76	0.33	8.00	4.05	0.12	0.7	13.07	<0.05
Sand	94.94	2.23	0.67	0.03	0.00	0.00	0.76	0.09	
Diatomite	80.5	7.86	2.37	0.19	0.18	0.05	0.72	0.12	

**Table 3 materials-17-00626-t003:** TGA results for pastes and mortars tested at 2, 7, and 28 days, indicating LOI at specific temperature steps (250 °C, 500 °C, and 950 °C).

Sample	Activating Conditions	2 Days			7 Days			28 Days		
		LOI 250 °C	LOI 500 °C	LOI 950 °C	LOI 250 °C	LOI 500 °C	LOI 950 °C	LOI 250 °C	LOI 500 °C	LOI 950 °C
Pastes	Solid Glass	1.55	0.96	0.42	1.64	1.32	0.76	1.72	1.99	0.60
	Solution glass	1.75	0.94	0.39	1.75	1.10	0.49	1.95	2.28	0.56
	Standard	1.18	0.90	0.24	1.89	1.15	0.35	1.99	2.24	0.56
	Solid diatomite	1.55	0.90	0.34	1.45	1.97	0.51	1.94	2.02	0.59
	Solution diatomite	1.50	0.92	0.11	1.59	1.69	0.45	1.84	2.21	0.57
	Solid sand	1.80	1.01	0.25	1.56	1.81	0.53	2.32	1.74	0.74
	Solution sand	1.72	1.06	0.30	1.60	1.85	0.74	1.86	2.35	0.81
Mortars	Solid sand	0.51	0.49	0.22	0.65	0.40	0.18	0.94	0.86	0.65
	Solution sand	0.57	0.08	0.14	0.52	0.52	0.15	0.70	0.46	0.43
	Solid glass	0.53	0.26	0.02	0.62	0.73	0.11	0.76	0.45	0.26
	Solution glass	0.57	0.23	0.22	0.63	0.46	0.09	0.82	0.57	0.26
	Solid diatomite	0.38	0.30	0.05	0.62	0.43	0.34	0.64	0.67	0.60
	Solution diatomite	0.46	0.21	0.11	0.44	0.59	0.27	0.83	0.57	0.25

**Table 4 materials-17-00626-t004:** Mean compressive strength of all tested mortars, and respective standard deviation at different hydration times.

Mortars Results		
Time of Hydration	Mean Compressive Strength (MPa)	STD Deviation
2 days	8.4	4.0
7 days	36.0	1.6
28 days	59.5	2.8

## Data Availability

Data are contained within the article.

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
