# Peer review of "The Use of Solid Sodium Silicate as Activator for an Amorphous Wollastonitic Hydraulic Binder"

_materials, 2024, doi:10.3390/ma17030626_

Round 1

Reviewer 1 Report

Comments and Suggestions for Authors

GENERAL COMMENTS

The paper focuses on the synthesis of Na2SiO3 powders using various raw materials (sand, glass, and diatomite) to use as activators of amorphous wollastonitic hydraulic binders. 

The work is interesting and merits publication. However, the discussion of the results must be better developed, particularly explaining the differences in the FTIR spectra obtained in relation to the composition of the raw materials, and the differences in the compressive strength results.

A few specific comments to improve the paper are given below.

SPECIFIC COMMENTS

The text is generally well written, but I suggest using spell checker to correct typos, e.g., line 46: “hydrualic" instead of “hydraulic.”

The introduction can be better developed – there are numerous studies on this subject and only a few are referenced.

I suggest renaming section 2.3. as e.g., Analysis of the activators 

2.5. This section is lacking details on the sample preparation, curing conditions, number of replicates. Idem for section 2.7. What type of sand was used, in which proportions, type of samples, curing conditions, etc.

Number of replicates and standard deviations are missing in the results.

Comments on the Quality of English Language

The text is generally well written, but I suggest using spell checker to correct typos, e.g., line 46: “hydrualic" instead of “hydraulic.”

Author Response

  • Reviewer 1

The paper focuses on the synthesis of Na2SiO3 powders using various raw materials (sand, glass, and diatomite) to use as activators of amorphous wollastonitic hydraulic binders. 

The work is interesting and merits publication. However, the discussion of the results must be better developed, particularly explaining the differences in the FTIR spectra obtained in relation to the composition of the raw materials the differences in the compressive strength results.

Answer: A detailed description of the FTIR spectra of the products was added and related to the composition of the raw materials. On the discussion and result section, further analysis of the compressive strength results was introduced, and 2 new references were added on the discussion [42,43]

A few specific comments to improve the paper are given below. 

SPECIFIC COMMENTS

  1. The text is generally well written, but I suggest using spell checker to correct typos, e.g., line 46: “hydrualic" instead of “hydraulic.”- Corrected

  1. The introduction can be better developed – there are numerous studies on this subject and only a few are referenced. A new paragraph and 9 new references were added.

  1. I suggest renaming section 2.3. as e.g., Analysis of the activators Corrected

  1. 5. This section is lacking details on the sample preparation, curing conditions, number of replicates. the following information was added: curing and mixing conditions, time at which the samples were demolded and number of samples tested per age.

  1. Idem for section 2.7. What type of sand was used, in which proportions, type of samples, curing conditions, etc.: in 2.5 the following information was added: mixing conditions, type of sand, time at which the samples were demolded, and number of samples tested per age.

  1. Number of replicates and standard deviations are missing in the results. Number of replicas was added, standard deviations of the mortar results was calculated.

Comments on the Quality of English Language

  1. The text is generally well written, but I suggest using spell checker to correct typos, e.g., line 46: “hydrualic" instead of “hydraulic. Corrected

Reviewer 2 Report

Comments and Suggestions for Authors

In this paper, it has been found that the use of sodium silicate solutions as activators for amorphous wollastonite hydraulic binders increases the cost of the final product and the overall complexity of its use. Therefore, this paper focuses on the utilization of cost-effective raw materials to enhance the fabrication of alkaline activators. Laboratory-produced activators were analyzed for chemical composition and physical properties, and the results indicated that the use of glass powder as a silica source could enhance the performance of the activators. Overall, there are many issues that need to be improved in the experimental design and analysis of the results of the paper. The problems are as follows.

(1) Detailed information on the preparation of the activator in the paper is inadequate, including the preparation method of the activator, the source of raw materials and the treatment process. It is suggested that the authors describe the experimental conditions and operation steps in detail so that readers can clearly understand the experimental design.

(2) In the experimental methodology, the authors mention the use of reactive paste specimens of specific dimensions and water-solid ratios, and compression strength tests. However, when describing the experimental steps, more details are needed on the specific steps of sample preparation and compression tests.

(3) For the naming and identification of peaks in Figures 2 and 3, can you provide more experimental data or references to support your conclusions?

(4) Have you considered other factors that may lead to unreacted silica in your samples?

(5) More detailed explanations and analyses need to be provided for the data on heat flow and cumulative heat.

(6) As far as I know, mixing time, temperature, etc. may affect the performance of the sample, have you fully considered the influencing factors?

(7) The authors mention the results of the compressive strength of different specimens, but these results are not analyzed and discussed in depth. It is suggested that the authors should analyze the comparative compressive strength of different specimens in the discussion section of the results and explain the possible reasons for the differences.

(8) The references mentioned by the authors do not seem to address the latest research findings and they are not adequately compared and discussed in the text. It is recommended that the authors add some of the latest relevant studies in the references section and provide an in-depth review of these relevant studies in order to better relate their findings to the current frontiers of the field.

(9) The authors mentioned some theoretical models related to water chemistry in the paper, but did not explain in detail the correlation between these models and their research results. It is suggested that the authors explain in detail the correlation between these theoretical models and the experimental data in the Discussion and Conclusions section to ensure the reliability and accuracy of the experimental results.

(10) Some of the trends in the experimental data are mentioned in the discussion of results section, but these trends are not explained and analyzed in depth. It is recommended that the authors provide a detailed explanation of the trends in the experimental data in the discussion of results section and explore possible mechanisms and influencing factors.

(11) The paper is loosely structured and it is recommended that more data explanations and mechanisms be added and recomposed.

Comments on the Quality of English Language

The language quality of the article is not a problem, but there are some formatting and font errors that need to be corrected.

Author Response

In this paper, it has been found that the use of sodium silicate solutions as activators for amorphous wollastonite hydraulic binders increases the cost of the final product and the overall complexity of its use. Therefore, this paper focuses on the utilization of cost-effective raw materials to enhance the fabrication of alkaline activators. Laboratory-produced activators were analyzed for chemical composition and physical properties, and the results indicated that the use of glass powder as a silica source could enhance the performance of the activators. Overall, there are many issues that need to be improved in the experimental design and analysis of the results of the paper. The problems are as follows.

  • Detailed information on the preparation of the activator in the paper is inadequate, including the preparation method of the activator, the source of raw materials and the treatment process. It is suggested that the authors describe the experimental conditions and operation steps in detail so that readers can clearly understand the experimental design.

Answer: The source of raw materials and the milling process of the silica-rich materials was added. The treatment of the product after its synthesis was also disclosed.

(2) In the experimental methodology, the authors mention the use of reactive paste specimens of specific dimensions and water-solid ratios, and compression strength tests. However, when describing the experimental steps, more details are needed on the specific steps of sample preparation and compression tests. Corrected

(3) For the naming and identification of peaks in Figures 2 and 3, can you provide more experimental data or references to support your conclusions?

Answer: A detail description of the FTIR spectra was added (Figure 2), and three new references were provided on the DRX interpretation (Figure 3).

(4) Have you considered other factors that may lead to unreacted silica in your samples?

Answer: samples that displayed unreacted silica were sand and the diatomite-based activator, both silica-rich raw materials with higher crystalline content than glass which is mostly amorphous. This difference may have affected the SiO2 solubility and, consequently, the reaction process.

(5) More detailed explanations and analyses need to be provided for the data on heat flow and cumulative heat.

Answer: Table 3 was removed to avoid confusion, and an interpretation of the calorimetric results was included.

(6) As far as I know, mixing time, temperature, etc. may affect the performance of the sample, have you fully considered the influencing factors?

Answer: Since all activators were synthetized with the same method, and the pastes and mortars were made and cured under the same conditions, the difference in performance can only be related with the only variable under study, the silica-rich raw material used for synthesizing the activator.

(7) The authors mention the results of the compressive strength of different specimens, but these results are not analyzed and discussed in depth. It is suggested that the authors should analyze the comparative compressive strength of different specimens in the discussion section of the results and explain the possible reasons for the differences.

Answer: On the discussion and result sections, further analysis of the compressive strength results was introduced.

(8) The references mentioned by the authors do not seem to address the latest research findings and they are not adequately compared and discussed in the text. It is recommended that the authors add some of the latest relevant studies in the references section and provide an in-depth review of these relevant studies in order to better relate their findings to the current frontiers of the field.

Answer: A new paragraph and 9 new references were added in the introduction.

(9) The authors mentioned some theoretical models related to water chemistry in the paper, but did not explain in detail the correlation between these models and their research results. It is suggested that the authors explain in detail the correlation between these theoretical models and the experimental data in the Discussion and Conclusions section to ensure the reliability and accuracy of the experimental results.

Answer: The model is explained in more detail and further analysis of the correlation was added in the discussion and the conclusion.

(10) Some of the trends in the experimental data are mentioned in the discussion of results section, but these trends are not explained and analyzed in depth. It is recommended that the authors provide a detailed explanation of the trends in the experimental data in the discussion of results section and explore possible mechanisms and influencing factors.

Answer: On the discussion and result sections, further analysis of the results was introduced.

(11) The paper is loosely structured and it is recommended that more data explanations and mechanisms be added and recomposed.

Answer: The structure of the paper was improved, data explanation and mechanisms hypothesis were also added in the results and discussion sections.

Reviewer 3 Report

Comments and Suggestions for Authors

The work of  M. Antunes, R. L. Santos, R. B. Horta and R. Colaço is very interesting and has an ecological orientation, but I have many remarks on the work:
1. line 116.
There is no reason for the description but FTIR and XRD methods to be in the same paragraph. These methods are very different in methodology, not similar in measurements, calculations, etc., so it is better to separate them in separate paragraphs with the corresponding description of conditions.
2. line 161
Under what experimental conditions was the heating conducted in the temperature range 250-500o C? Describe the methodology and how mass losses were calculated. Are there results that can be presented graphically or tabulated?
3. line 170-173
The interpretation of the CO3 bands is presented in an unprofessional manner. The authors use the term "peak", whereas for IR spectroscopy it is customary to use the term "band". The bands correspond to vibrational and deformational vibrations of the anions in the structure of the compounds. The results of IR spectroscopy do not give a direct result for phase identification. The results give the vibrations of the functional groups, and from the character, location, intensity and other features, inferences are made about the assignments to the corresponding functional groups in the particular compounds. Therefore, assigning the bands represented by the IR spectra in Fig. 2 to the spectra of a given compound is a methodological error.
4. line 194
Isothermal Calorimetry results are presented in Fig. 4 and in Table 3. The results are not comparable, there is no explanation of how the values in column 5 of Table 3 were obtained. These are not direct measurement results but the result of calculations and therefore it is necessary to explain how they are obtained.
Conclusion
The work is interesting, but there are many terminological inaccuracies, such as
- FTIR measurements - the results refer directly to the identification of compounds, not to vibrational and strain fluctuations of functional groups.
- Thermogravimetric data are missing, only final results are presented without methodology of measurements, calculations, etc.
- Line 226 - Reference No. 18 is misquoted
- Line 251 - NaOH is presented as a salt, and this is an alkali metal hydroxide.
The work requires methodological and terminological refinement. It is not suitable for publication in its present form.

Author Response

The work of  M. Antunes, R. L. Santos, R. B. Horta and R. Colaço is very interesting and has an ecological orientation, but I have many remarks on the work:

  1. line 116.
    There is no reason for the description but FTIR and XRD methods to be in the same paragraph. These methods are very different in methodology, not similar in measurements, calculations, etc., so it is better to separate them in separate paragraphs with the corresponding description of conditions. Corrected
  2. line 161
    Under what experimental conditions was the heating conducted in the temperature range 250-500o C? Describe the methodology and how mass losses were calculated. Are there results that can be presented graphically or tabulated?

Answer: The experimental condition of the TGA technique were disclosed and the tabulated results obtained were added (table 3)

  1. line 170-173
    The interpretation of the CO3 bands is presented in an unprofessional manner. The authors use the term "peak", whereas for IR spectroscopy it is customary to use the term "band". The bands correspond to vibrational and deformational vibrations of the anions in the structure of the compounds. The results of IR spectroscopy do not give a direct result for phase identification. The results give the vibrations of the functional groups, and from the character, location, intensity and other features, inferences are made about the assignments to the corresponding functional groups in the particular compounds. Therefore, assigning the bands represented by the IR spectra in Fig. 2 to the spectra of a given compound is a methodological error.

Answer: The term and the assigning of the band were corrected. 

  1. line 194
    Isothermal Calorimetry results are presented in Fig. 4 and in Table 3. The results are not comparable, there is no explanation of how the values in column 5 of Table 3 were obtained. These are not direct measurement results but the result of calculations and therefore it is necessary to explain how they are obtained.

Answer: Table 3 was removed to avoid confusion, and an interpretation of the calorimetric results was included.

Conclusion
The work is interesting, but there are many terminological inaccuracies, such as
- FTIR measurements - the results refer directly to the identification of compounds, not to vibrational and strain fluctuations of functional groups. Corrected
- Thermogravimetric data are missing, only final results are presented without methodology of measurements, calculations, etc. Corrected
- Line 226 - Reference No. 18 is misquoted Corrected
- Line 251 - NaOH is presented as a salt, and this is an alkali metal hydroxide. Corrected
The work requires methodological and terminological refinement. It is not suitable for publication in its present form.

Round 2

Reviewer 1 Report

Comments and Suggestions for Authors

The authors have revised the document taking into account the reviewers’ suggestions. However, there are still a few minor issues that require revision, as specified below.

Section 2.8: Please, specify the number of replicates used for the mechanical strength tests.

The graph in Figure 5 is missing standard deviations. Idem for Fig. 6.

The caption in Figure 6 also contains Fig. 5 caption.

Comments on the Quality of English Language

The English language still requires a general revision to avoid typos (e.g., line 418: “lowe” instead of “low”).

Author Response

Reviewer 1

Comments and Suggestions for Authors

The authors have revised the document taking into account the reviewers’ suggestions. However, there are still a few minor issues that require revision, as specified below.

  • Section 2.8: Please, specify the number of replicates used for the mechanical strength tests.
  • The following information was added Finally, one replica per age was tested, which included one flexural test and two compressive strength tests after 2, 7, and 28 days of hydration in an Ibertest Autotest 400/10 instrument 
  •  
  • The graph in Figure 5 is missing standard deviations. Idem for Fig. 6.
  • Ony one replica per age was tested due to the difficulty in material production
  •  
  • The caption in Figure 6 also contains Fig. 5 caption.
  • corrected

Comments on the Quality of English Language

  • The English language still requires a general revision to avoid typos (e.g., line 418: “lowe” instead of “low”).
  • Corrected

Reviewer 2 Report

Comments and Suggestions for Authors

The manuscript was modified in a reasonable manner. The paper can be accepted.

Comments on the Quality of English Language

Over all, the English is not bad.

Author Response

English revision was made

Reviewer 3 Report

Comments and Suggestions for Authors

The authors have taken note of the comments and improved the quality of the work.
I leave the final decision to the editors of the journal to judge the appropriateness of publishing the material

Author Response

The authors have further improved the quality of the manuscript